# Heat-Inactivation of Fetal and Newborn Sera Did Not Impair the Expansion and Scaffold Engineering Potentials of Fibroblasts

**DOI:** 10.3390/bioengineering8110184

**Published:** 2021-11-13

**Authors:** Félix-Antoine Pellerin, Christophe Caneparo, Ève Pellerin, Stéphane Chabaud, Martin Pelletier, Stéphane Bolduc

**Affiliations:** 1Department of Microbiology, Faculté de Sciences et Génie, Université Laval, Québec, QC G1V 0A6, Canada; felix-antoine.pellerin.1@ulaval.ca; 2Centre de Recherche en Organogénèse Expérimentale/LOEX, Regenerative Medicine Division, CHU de Québec-Université Laval Research Center, Québec, QC G1J 1Z4, Canada; christophe.caneparo@crchudequebec.ulaval.ca (C.C.); eve.pellerin.1@ulaval.ca (È.P.); Stephane.Chabaud@crchudequebec.ulaval.ca (S.C.); 3Infectious and Immune Disease Division, CHU de Québec-Université Laval Research Center, Québec, QC G1V 0A6, Canada; martin.pelletier@crchudequebec.ulaval.ca; 4Department of Microbiology-Infectious Diseases and Immunology, Faculty of Medicine, Laval University, Québec, QC G1V 0A6, Canada; 5ARThrite Research Center, Laval University, Québec, QC G1V 4G2, Canada; 6Division of Urology, Department of Surgery, CHU de Québec-Université Laval, Québec, QC G1V 4G2, Canada

**Keywords:** serum, heat-inactivation, 2D cell culture, metabolism, tissue engineering, mechanical properties

## Abstract

Heat inactivation of bovine sera is routinely performed in cell culture laboratories. Nevertheless, it remains debatable whether it is still necessary due to the improvement of the production process of bovine sera. Do the benefits balance the loss of many proteins, such as hormones and growth factors, that are very useful for cell culture? This is even truer in the case of tissue engineering, the processes of which is often very demanding. This balance is examined here, from nine populations of fibroblasts originating from three different organs, by comparing the capacity of adhesion and proliferation of cells, their metabolism, and the capacity to produce the stroma; their histological appearance, thickness, and mechanical properties were also evaluated. Overall, serum inactivation does not appear to provide a significant benefit.

## 1. Introduction

Since the second half of the 20th century, molecular and cell biology has been based on the culture of cell lines or primary cell populations. The proliferation and maintenance of the cells are mainly achieved through a cell culture medium containing serum. Fetal bovine serum (FBS) is currently considered a reference even if almost everyone agrees on replacing it with less expensive and more ethical sources that would allow more reproducible results [1,2]. Nevertheless, before its use, FBS is routinely heat-inactivated, generally at 56 °C for 30 min, mainly to prevent the action of complement components and contamination by mycoplasma [3,4]. Part of the growth factors, vitamins, and precious nutrients required for cell culture and activity can be lost during this exposition to heat. From the beginning of cell culture, there is a gap in technologies to assure the quality of the products provided for cell culture. This is also true for the FBS production, storage, and quality assessment [1]. As early as 1980, a study demonstrated that complement components are present in very low or undetectable levels in FBS samples [3].

However, newborn calf serum (NBCS) or adult sera are also used and show higher concentrations of complement components than FBS [5]. Elimination of mycoplasma can also be achieved by an adequate filtration process [1]. Heat inactivation can positively affect experimental results, as demonstrated in the functions of human T cells [6], even though other studies state that heat inactivation is not required for lymphocyte functions [7]. The heat inactivation process can also create a bias and have a detrimental effect on several processes, such as cell attachment [8], neurite growth [9], proliferation, and metabolic activity of mesenchymal stromal cells, their stem/progenitor cell maintenance, and their ability for osteo-differentiation [10] or bovine embryo development in vitro [11]. In addition to cell culture, another potentially affected application can be immuno-analysis of viral antibodies. Heat inactivation presents pros or cons depending on the virus and the technique, favorable in the case of the West Nile virus [12] but unfavorable in the case of Human T-Cell Lymphotropic Virus Type III [13] or SARS-CoV-2 [14].

More recently, biology research has gone a step forward by producing reconstructed 3D tissues for implantation or to serve as research models through tissue engineering [15]. This technology relies mainly on scaffolds and cells cultivated with sera of various origins to produce tissues. It is legitimate to ask whether the demanding cell culture conditions required for tissue engineering would not benefit from the conservation of the various factors lost during the inactivation of the serum. Tissue engineering mainly relies on the use of pre-existing scaffolds of biomaterials seeded with cells. This concept is interesting because the production is generally fast and relatively inexpensive. However, the pre-existing structure can induce bias, such as cell misdifferentiation, discrepency from native scaffold composition and organization, etc.

Dr. François A. Auger then imagined a different approach. The first step was the demonstration by Switzer and Summer of the ability of human fibroblasts cultivated in the presence of ascorbic acid to produce collagen in 1972 [16]. This finding was followed by the discovery by Hata and Senoo in 1989, that these cells deposited enough matrix within a few days to create a three-dimensional (3D) tissue-like sheet [17]. On this basis, Dr. François A. Auger developed, in 1998, a method for blood vessel reconstruction without the need for exogenous biomaterials [18,19]. This method was called “self-assembly.” In this technique, cells build their microenvironment through matrix deposition. It can be expected that the tissues reconstructed by this technique recreate more faithfully the interactions between cells and the extracellular matrix (ECM) and those between cells themselves compared to what can be obtained using the other tissue engineering models. The scope of this approach grew over time to allow the production of effective tissues for organ transplantation [20] and as models for fundamental studies [2,21,22].

The study presented here evaluates the potential of FBS and NBCS in their untreated or heat-inactivated form for the cell culture and reconstruction of the dermis, lamina propria (bladder submucosa), and vaginal stroma. In addition, adhesion, proliferation, and metabolism of cells cultivated in 2D and histology and mechanical properties of reconstructed 3D tissues were assessed.

## 2. Materials and Methods

### 2.1. Ethics Statement

All procedures involving patients were conducted according to the Helsinki Declaration and were approved by the Research Ethical Committee of CHU de Québec-Université Laval. Donors’ consent for tissue harvesting was obtained for each specimen. Experimental procedures were performed in compliance with the CHU de Québec guidelines.

### 2.2. Cell Culture

Human skin, bladder, and vaginal mucosa biopsies were obtained from patients undergoing plastic surgeries or benign pathologies. Three biopsies of each tissue were obtained, and fibroblasts and epithelial cells were isolated. The biopsy was rinsed, cut into small strips, and incubated overnight at 4 °C in a 500 mg/mL thermolysin solution (Sigma-Aldrich, Oakville, ON, Canada) diluted in a 4-(2-hydroxyethyl)-1piperazineethanesulfonic acid (HEPES, MP Biomedicals, Montreal, QC, Canada) buffer with 1 mM of CaCl_2_ pH 7.4 (Sigma-Aldrich). The epithelium was gently separated from the stroma with forceps. The stroma was incubated for 3 h in a 125 U/mL collagenase H solution (Boehringer Mannheim, Laval, QC, Canada) diluted in Dulbecco-Vogt modification of Eagle’s medium (DMEM, Invitrogen, Burlington, ON, Canada), supplemented with 10% fetal bovine serum (FBS, Avantor Seradigm, Radnor, PA, USA), 100 U/mL penicillin (Sigma-Aldrich, Oakville, ON, Canada), and 25 µg/mL gentamicin (Schering, Pointe-Claire, QC, Canada) at 37 °C. Fibroblasts (DF1, DF2, and DF3 from skin dermis; BF1, BF2, and BF3 from bladder lamina propria; and VF1, VF2, and VF3 from vagina) were grown in DMEM with appropriate serum (depending on the condition) and antibiotics. They were used between passages 3 to 6.

### 2.3. Serum Inactivation

Sera used in this study were: untreated FBS, heat-inactivated FBS, untreated newborn calf serum (NBCS, Gibco Life Technologies, New-Zealand), and heat-inactivated NBCS. Heat inactivation followed the standard procedures. Briefly, serum bottles were thawed overnight in a beaker containing water in a fridge at 4 °C to obtain a more homogenous temperature change. Serum bottles were then placed in a water bath at 56 °C for 30 min. Serum was aliquoted in 50 mL sterile tubes and placed overnight at −20 °C before being stored at −80 °C until used. Control sera (untreated) followed the same procedure except for the incubation in the 56 °C bath. Instead, they stayed at 4 °C.

### 2.4. Adhesion and Proliferation of Fibroblasts

After 3 weeks of cell culture in DMEM with appropriate serum (FBS, heat-inactivated FBS, untreated NBCS, and heat-inactivated NBCS), fibroblasts were seeded at 10% confluence in 12-well plates (5000 Fibroblasts/cm^2^, in 1 mL cell culture medium/well). Appropriate medium (DMEM + serum of the condition tested) was changed daily with fresh medium for 3 days. Each day, cells from three wells were harvested and countered separately using a Coulter-Beckmann Z2 system. A proliferation curve was performed to calculate the doubling time. The regression curve formula extrapolated from the proliferation curve was P = Po × e^gx^, where P was the number of cells on day x, Po was the cell number at day 0, and g was the exponential coefficient. Doubling time was calculated with the following formula: D = ln2/g (https://en.wikipedia.org/wiki/Doubling_time, accessed on 5 November 2021). For the adhesion measurement, the cell count of the indicated condition on day one was compared to the cell count on day one for the reference (i.e., heat-inactivated FBS) except for the DF1 population, where untreated FBS served as reference.

### 2.5. Metabolism Evaluation of Fibroblast Cultures

Fibroblasts (200,000 cells/cm^2^, in 0.1 mL cell culture medium/well) were seeded in XFe96 96-well plates. Cells were incubated for 3 days with DMEM in the presence of the appropriate serum (FBS, heat-inactivated FBS, untreated NBCS, and heat-inactivated NBCS) before analysis. Seahorse XFe96 sensor cartridge plates (Agilent/Seahorse Bioscience, Santa Clara, CA, USA) were hydrated the day before the analysis with the XF Calibrant (Agilent/Seahorse Bioscience, Santa Clara, CA, USA) and incubated at 37 °C without CO_2_ overnight. Before the energy metabolism measurements, cells were washed and incubated for 1 h with Glyco Stress media or Mito Stress media. Glyco Stress media contained XF Base Medium (minimal DMEM) (Agilent/Seahorse Bioscience, Santa Clara, CA, USA), supplemented with 2 mM L-glutamine (Wisent Bioproducts Inc., Saint-Jean-Baptiste, QC, Canada). Mito Stress media contained XF Base Medium (Agilent/Seahorse Bioscience, Santa Clara, CA, USA), supplemented with 2 mM L-glutamine (Wisent Bioproducts Inc., Saint-Jean-Baptiste, QC, Canada), 1 mM sodium pyruvate (Wisent Bioproducts Inc., Saint-Jean-Baptiste, QC, Canada), and 10 mM D-(+)-glucose (Millipore Sigma, Oakville, ON, Canada). Extracellular acidification rate (ECAR), representative of glycolytic metabolism, and oxygen consumption rate (OCR), representative of mitochondrial respiration, were determined using the XFe Extracellular Flux Analyzer (Agilent/Seahorse Bioscience, Santa Clara, CA, USA). The mitochondrial respiration was established by the sequential injection of the ATP synthase inhibitor oligomycin 1.5 µM (67.5% oligomycin A complex) (Cayman Chemical, Ann Arbor, MI, USA), the mitochondrial uncoupler trifluoromethoxy carbonylcyanide phenylhydrazone (FCCP) 0.5 µM (Cayman Chemical, Ann Arbor, MI, USA), and a combination of the mitochondrial complex I inhibitor rotenone 0.5 µM (MP Biomedicals, Santa Ana, CA, USA) and mitochondrial complex III inhibitor antimycin A 0.5 µM (Millipore Sigma, Oakville, ON, Canada). The glycolytic metabolism was established by the sequential injection of D-(+)-glucose 10 mM (Millipore Sigma, Oakville, ON, Canada), the ATP synthase inhibitor oligomycin 1.5 µM (67.5% oligomycin A complex) (Cayman Chemical, Ann Arbor, MI, USA), to inhibit mitochondrial respiration and force the cells to maximize their glycolytic capacity, and 2-deoxy-D-glucose (2-DG) 50 mM (Alfa Aesar, Ward Hill, MA, USA), a competitive inhibitor of glucose. At least three measurement cycles (3 min of mixing + 3 min of measuring) were completed before and after each injection. OCR and ECAR were calculated using the Wave software v2.6 (Agilent/Seahorse Bioscience, Santa Clara, CA, USA). Basal and maximal mitochondrial and glycolysis corresponded to the measure indicated by the black arrow, basal measure, and the grey arrow maximal measure (Appendix A Appendix A) [23]. Energy metabolism was normalized according to the number of cells using the CyQuant Cell proliferation assay kit (Invitrogen, Burlington, ON, Canada) and following the manufacturer’s instructions. The fluorescence of each well was measured at 485 nm/535 nm during 0.1 s using the Victor2 1420 MultiLabel Counter plate reader (Perkin Elmer Life Sciences, Waltham, MA, USA) and Wallac 1420 software (PerkinElmer, Waltham, MA, USA). The normalization values were calculated from the fluorescence measurements with Microsoft Excel software (Microsoft, Redmond, WA, USA) and applied to the metabolic values.

### 2.6. Stroma Reconstruction Using the Self-Assembly Approach

Fibroblasts at passage three were seeded to confluence (40,000 cells/cm^2^, in 2 mL cell culture medium/well) in six-well plates, including a paper anchorage device, weighed down by four small polished stainless-steel lingots (0.6 g each), and cultured in DMEM plus serum (depending on the condition), supplemented with 50 µg/mL ascorbate for 14 days. Fibroblasts were reseeded on top of the initial stromal sheet to form a confluent layer on its upper surface (400,000 fibroblasts/cm^2^, in 2 mL cell culture medium/well). The culture was continued for 14 additional days. Two fibroblast sheets were superimposed without interposed air bubbles and clipped with three surgical ligaclips (Ethicon Endo Surgery, Cincinnati, OH, USA). Three large lingots (12 g) were put on top of the stroma to help sheet fusion. A surgical sponge (Merocel, Mystic, CT, USA) was placed between the large lingots and the stroma to decrease the effect of the mechanical load. The sponge and the large lingots were removed after 24 h. The stroma was cultured for 3 additional days to allow sheet fusion. In the case of the lamina propria (bladder stroma), a mix of 10% dermal fibroblasts and 90% bladder fibroblasts was used. A total of 144 tissues were produced: four tissues by condition (untreated FBS, heat-inactivated FBS, untreated NBCS, and heat-inactivated NBCS) for each of the nine fibroblast populations.

### 2.7. Histology and Thickness Measurement

Sections of each sample (four stromae for each condition) were fixed in Histochoice tissue fixative (Amresco, Solon, OH, USA) and embedded in paraffin. Histological sections 5 µm thick were cut and stained using Masson’s trichrome (MT). The thickness of the stroma was assessed using a Zeiss Axio Imager M2 microscope equipped with an AxioCam HR Rev3 camera (Carl Zeiss, Oberkochen, Germany). Images were processed with the AxioVision 40 V4.8.2.0software (Carl Zeiss, Oberkochen, Germany), and scale bars were added with ImageJ software (NIH, Bethesda, MD, USA). Images were analyzed using ImageJ software. A total of nine measurements were made on three MT-stained images (×40 magnification) of reconstructed tissues.

### 2.8. Mechanical Testings

Mechanical properties of the reconstructed stromae were assessed by uniaxial tensile testing using an ElectroPuls E1000 mechanical tester (Instron Norwood, MA, USA). Bone-shaped biopsies were cut with a designed stainless-steel punch. Both extremities of the specimen were stretched at a constant rate of 0.2 mm/s until the tissue ruptured. Data were analyzed using Minitab (Minitab, State College, PA, USA) to provide maximal strength, ultimate tensile strength, elastic modulus, and failure strain. For each condition, four stromae were tested. All data were expressed as mean ± standard deviation, and the graphics were generated using Microsoft Office Excel.

### 2.9. Statistic

Graphical representation and statistical analyses were performed using GraphPad Prism v.9.2 Software (San Diego, CA, USA). Dots present all the values for every experiment, except metabolism measurements, where the results are expressed as mean ± standard error of the mean (SEM). Statistical analyses were performed using one-way analysis of variance (ANOVA), followed by a comparison of conditions. Statistical significance was established at *p* < 0.05.

## 3. Results

### 3.1. Adhesion in 2D Cultures

In cell culture, the first step is the adhesion of the cells on a surface. This surface is often the tissue culture plastic (TCP, cell-culture treated polystyrene). The adhesion of the nine cell populations, three dermal fibroblast populations (DFs) isolated from skin dermis, three bladder fibroblast populations (BFs) isolated from bladder lamina propria, and three vaginal fibroblast populations (VFs) isolated from vaginal submucosa, was evaluated. The adhesion of DFs, BFs, and VFs remained unchanged regardless of its untreated vs. inactivation status (Figure 1A). The only significant difference was obtained when the heat-inactivated serum conditions were compared with a lower adhesion of VF when NBCS was used compared to the condition with FBS. Results for individual fibroblast populations can be found in Appendix A.

### 3.2. Proliferation in 2D Cultures

Once cells have adhered to the culture surface, their proliferation is required to amplify the cell population to have enough cells to be used to reconstruct a tissue. This parameter was evaluated in Figure 1B. Proliferation (inverse of doubling time measurement) of DFs, BFs, and VFs remained unchanged regardless of serum type or heat-inactivation status. Results for individual fibroblast populations can be found in Appendix A.

### 3.3. Mitochondrial Respiration in 2D Cultures 

Mitochondrial respiration is the primary source of energy for a cell. The oxygen consumption rate (OCR) was measured in real-time using an extracellular flux analyzer. Mitochondrial respiration of the DFs, BFs, and VFs was not impacted by serum type or serum heat-inactivation (Figure 2). Results for individual fibroblast populations can be found in Appendix A.

### 3.4. Glycolysis in 2D Cultures

Glycolysis can serve to provide additional energy to the cells during a very demanding process. However, glycolysis also produces metabolites, such as lactate, which can unfavorably impact healthy cell culture. Therefore, the extracellular acidification rate (ECAR) was measured in real-time using an extracellular flux analyzer. Glycolysis of the DFs, BFs, and VFs was not impacted by serum type or serum heat-inactivation (Figure 3). Results for individual fibroblast populations can be found in Appendix A.

### 3.5. Histology of 3D Reconstructed Tissues

The histologic aspect of the tissue was examined on Masson trichrome stained slices (Figure 4). Fusion of the two reconstructed sheets to form a cohesive stroma and homogenous distribution of the ECM was expected. All reconstructed stroma showed a cohesive stroma consisting of ECM and cells. Fusion of the two stacked sheets was achieved. From a histological point of view, the type of serum or the heat-inactivation of serum did not impact the gross organization of the stroma. Representative histology pictures for stromae produced from individual fibroblast populations can be found in Appendix A.

### 3.6. Thickness of 3D Reconstructed Tissues

The thickness of a 3D tissue reconstructed by the self-assembly technique is an insight into the mechanical properties. The ECM can depend on the organ-specific origin of the cells, but the accumulation of ECM generally correlates with the strength of the tissue. This parameter was measured on histological slices (Figure 5). The thickness of the stroma reconstructed using DFs, BFs, and VFs was not impacted by serum type or serum heat-inactivation. Results for individual fibroblast populations can be found in Appendix A.

### 3.7. Maximal Strength of 3D Reconstructed Tissues

The maximal strength obtained by uniaxial tensile testing allows the evaluation of the tissue’s ability to resist manipulation by surgeons and, in part, the resistance to sutures. Results from mechanical testing to measure the maximal strength are presented in Figure 6A. The maximal strength of the stroma reconstructed using DFs and VFs remained unchanged. The tissues were produced using BFs with FBS, where untreated serum condition showed greater maximal strength than those produced using the heat-inactivated serum. Additionally, untreated FBS allowed the production of tissue using BFs with greater maximal strength than the tissues made with untreated NBCS. Results for individual fibroblast populations can be found in Appendix A.

### 3.8. Elastic Modulus of 3D Reconstructed Tissues

The elasticity of the skin, bladder, and vagina is essential for their function; therefore, the elastic modulus should be as low as possible in inverse relation to elasticity. The elastic modulus is the slope of the strength as a stress function. Results from mechanical testing to measure the elastic modulus are presented in Figure 6B. Elastic modulus of the stroma reconstructed using DFs and VFs remained unchanged. Untreated FBS allowed the production of tissue using BFs with a greater elastic modulus than the tissue produced with untreated NBCS. Results for individual fibroblast populations can be found in Appendix A.

### 3.9. Ultimate Tensile Strength of 3D Reconstructed Tissues

The ultimate tensile strength (UTS), the intrinsic strength of the material, is helpful in biomaterial engineering when the amount of material can be increased or decreased to produce tissues with the desired mechanical properties. However, in the case of the tissues produced using the self-assembly method, its accuracy is more disputable. Of course, it is always possible to vary the number of stacked sheets to obtain greater mechanical resistance, but this also impacts other properties, such as elasticity. Nevertheless, its main interest remains that it allows comparison with biomaterials used in published works. Values are shown in Figure 7A. UTS of the stroma reconstructed using DFs and VFs remained unchanged. Untreated FBS allowed the production of tissue using BFs with a greater UTS than the tissue produced with untreated NBCS. Results for individual fibroblast populations can be found in Appendix A.

### 3.10. Failure Strain of 3D Reconstructed Tissues

The failure strain measures the deformability of the tissue, i.e., the elongation of the tissue when it breaks (Figure 7B). No difference was observed when DFs, BFs, or VFs were used, regardless of the heat-inactivation status or sera type. Results for individual fibroblast populations can be found in Appendix A.

## 4. Discussion

Heat-inactivation of sera can be time-consuming, especially in laboratories that use large amounts of sera, such as tissue engineering facilities. Therefore, the usefulness of such a procedure must be questioned, especially as the historical arguments used to support it are now obsolete. However, cell culture, and particularly the reconstruction of tissues from cells, is still not understood, and many parameters are involved. In particular, it has been established that cells originating from different patients or extracted by different protocols can give opposite results under the same experimental conditions. Additionally, cells belonging to the same family (for example, fibroblasts), but coming from different organs, respond differently, retaining the characteristics of their organ of origin in culture.

The production of tissues from a patient’s cells involves two main steps: (1) the extraction and amplification; (2) the reconstruction of the 3D tissues. Adhesion, proliferation, and metabolism of nine fibroblast populations from three different organs were evaluated in 2D cell cultures. Fibroblasts from the skin, bladder, and vagina were chosen because the reconstruction of substitutes of these source organs using the self-assembly method that has been published [22,24,25,26]. The diversity of primary fibroblast populations in terms of organ type, anatomical sites, sex, and age constitutes one of the limitations of this article. Nevertheless, several other cells that are usual for tissue-engineering reconstruction, such as keratocytes (fibroblasts from the cornea) [27] or adipose-derived stromal/stem cells [28] used for fat or bone reconstruction, as well as epithelial cells, could be used in further studies.

The heat-inactivation of the sera affected neither the adhesion ability of the cells (Figure 1A) nor their proliferation (Figure 1B). No significant modification of the metabolic activity of the cells was found, and differences remained anecdotal (Figure 2 and Figure 3). 

The same cell populations were used to reconstruct scaffolds, free of exogenous biomaterials, using the self-assembly approach. All reconstructed stromae were well fused and cohesive. The organization of the ECM also appeared to be similar between the different conditions (Figure 4). However, the thickness of some stromae seemed sometimes different (Appendix A), even if no statistically significant differences were found (Figure 5). When using FBS, four cell populations gave thicker tissues when serum remained untreated, and two were thicker using heat-inactivated FBS. For NBCS, three cell populations gave thicker tissue with untreated serum and two cell populations did the same when using heat-inactivated serum (Appendix A). It can be noted that, generally, the heat-inactivation had few effects on tissue thickness (Figure 5). The maximal strength was also mildly affected by serum treatment, especially with BFs, which produced stronger tissues when FBS remained untreated (Figure 6A). Three cell populations showed increased mechanical strength with U-FBS (Appendix A). The elasticity, the UTS, and the ability to be deformed were also mildly affected (Appendix A). Globally, our results demonstrated that heat-inactivation of sera had little or no effect on various parameters in 2D and 3D cell cultures. Interestingly, the BF-derived tissues produced using NBCS were weaker (Figure 6A and Figure 7A). Combined with the previously obtained results for the minimal differentiation of urothelial cells on the top of such a stroma [24], NBCS should be avoided to reconstruct urologic tissues. Such a result could be, in part, explained by the nature of BFs. In a previous study, the scaffold formed by the bladder fibroblasts showed spots of degraded ECM with increasing MMP activity, resulting in a weakened tissue, which is not observed using dermal fibroblasts [29]. One explanation for such a discrepancy in tissue properties between substitutes derived from bladder and dermal fibroblasts may be the nature of bladder fibroblasts. Several studies illustrate the plasticity of bladder fibroblasts, which have a phenotype varying from contractile muscle cells and secretory fibroblasts. High concentrations of Sonic Hedgehog (Shh) induce a fibroblast-like phenotype for bladder fibroblasts that secrete ECM to form the lamina propria and, in their turn, secrete the Bone Morphogenetic Protein 4 (BMP4). This factor, combined with the reduced concentration of Shh along the gradient, induces the smooth-muscle phenotype for the mesenchymal bladder cells in the area of what should become the detrusor [30,31]. It is possible that some unfavorable factors (in regard to the ECM secretion) remain in U-NBCS vs. U-FBS, or, conversely, some favorable factors are not lost in U-FBS but rather in U-NBCS.

However, it should not be concluded that the step of heat-inactivation of the serum is unnecessary. As it was seen, some cell populations seemed to be more affected than others. Equally and more importantly, the purpose of the reconstructed tissues must be questioned. Indeed, it seems evident that other parameters should be evaluated, for example, before proceeding with a transplant in a patient to see if the absence of heat-inactivation of the serum could trigger a rejection of the transplant [32,33,34] and possibly other negative consequences for a patient. If tissues are to be used as a model for research, it is also evident that studies involving the immune system [35,36] could be positively or negatively impacted. Studies should be carried out to study the impact of omitting heat-inactivation of the serum on the epithelium differentiation and the preservation of their stem cells on the composition and the fine organization of the ECM, which can be very important when studying pathologies, such as fibrosis [37,38] or cancer [21,39].

## 5. Conclusions

The primary efforts of the tissue engineering community should be directed towards the development of chemically-defined and serum-free media, in which all parameters are effectively controlled. Indeed, the serum induces numerous biases and requires constant testing and revalidation of a new batch of serum to characterize it as it is highly variable. Additionally, in the context of personalized medicine, which should become more and more significant in the future, it is interesting to see that cells, even from the same organ, react quite differently to stimuli such as serum and its modifications.

## Figures and Tables

**Figure 1 bioengineering-08-00184-f001:**
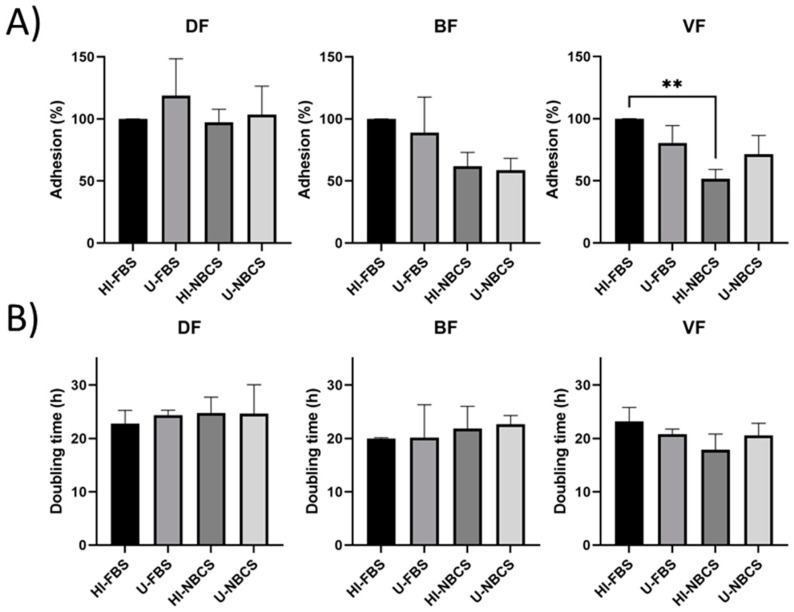
Adhesion and proliferation of fibroblasts primary populations depending on serum type and heat-inactivation status. DF is for fibroblast populations derived from skin dermis, BF is for fibroblast populations derived from bladder lamina propria, and VF is for fibroblast populations derived from vaginal stromae. HI- is for heat-inactivated, U- is for untreated, FBS is for fetal bovine serum, and NBCS is for newborn calf serum. Asterisks (*) illustrate significant differences. Two asterisks illustrate a *p*-value between 0.01 and 0.005. (**A**) Adhesion of fibroblast cell populations on a plastic Petri dish. The bars represent the mean ± standard deviation of the cell count 24 h after seeding normalized to the cell count obtained in HI-FBS. (**B**) Doubling time of fibroblast cell population calculated over a 3-day period. The bars represent the mean ± standard deviation of the doubling time of the cell population calculated from a growth curve established over 3 days. N = 3 independent fibroblast populations for each organ, *n* = 3 replicates.

**Figure 2 bioengineering-08-00184-f002:**
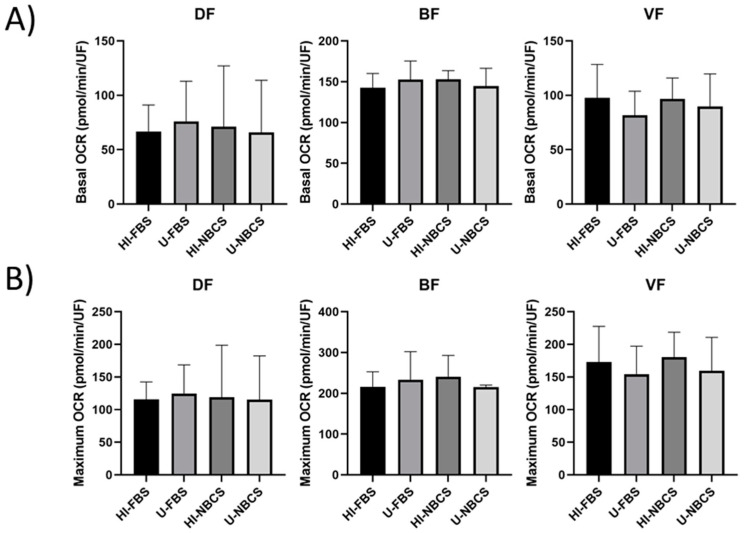
Mitochondrial respiration of fibroblasts’ primary populations depending on serum type and heat-inactivation status. DF is for fibroblast populations derived from skin dermis, BF is for fibroblast populations derived from bladder lamina propria, and VF is for fibroblast populations derived from vaginal stromae. HI- is for heat-inactivated, U- is for untreated, FBS us for fetal bovine serum, and NBCS is for newborn calf serum. Basal mitochondrial activity (**A**) and maximal mitochondrial activity (**B**) were measured by a real-time extracellular flux analyzer. The bars represent the mean ± standard deviation of oxygen consumption rate (OCR). N = 3 independent fibroblast populations for each organ, *n* = 4 replicates.

**Figure 3 bioengineering-08-00184-f003:**
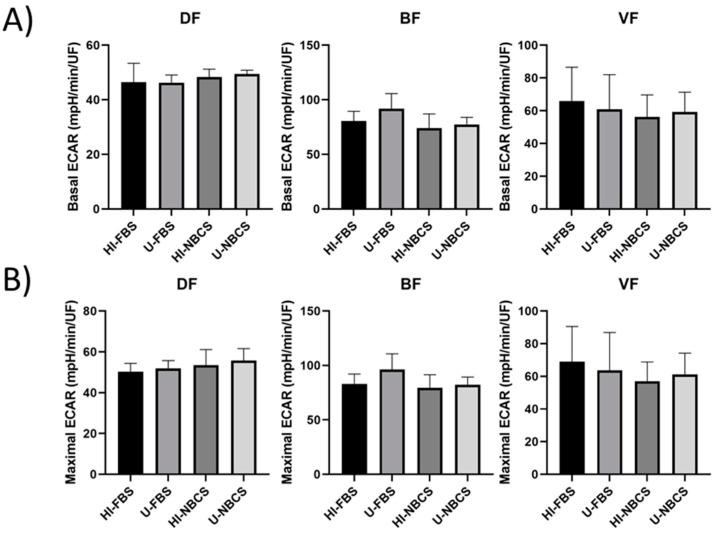
Glycolytic activity of fibroblasts primary populations depending on serum type and heat-inactivation status. DF is for fibroblast populations derived from skin dermis, BF is for fibroblast populations derived from bladder lamina propria, and VF is for fibroblast populations derived from vaginal stromae. HI- is for heat-inactivated, U- is for untreated, FBS is for fetal bovine serum, and NBCS is for newborn calf serum. Basal glycolytic activity (**A**) and maximal glycolytic activity (**B**) were measured by a real-time extracellular flux analyzer. The bars represent the mean ± standard deviation of the extracellular acidification rate (ECAR). N = 3 independent fibroblast populations for each organ, *n* = 4 replicates.

**Figure 4 bioengineering-08-00184-f004:**
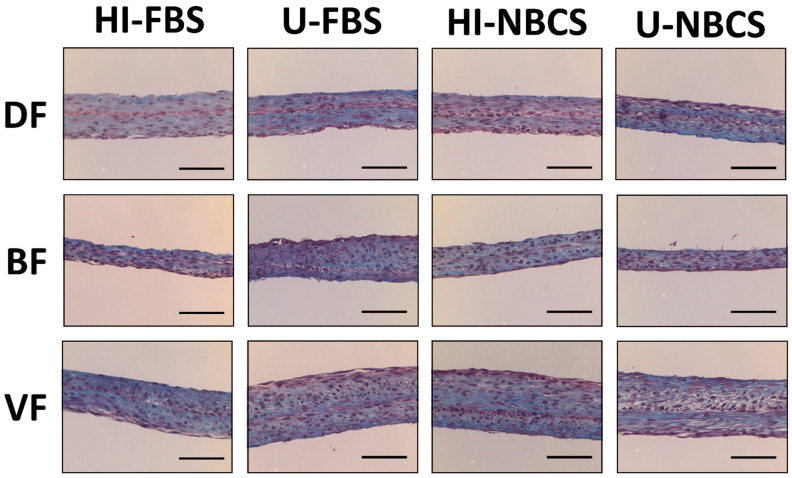
Histological aspect of reconstructed stromae by fibroblasts’ primary populations depending on serum type and heat-inactivation status stained with Masson’s Trichrome protocol. DF is for fibroblast populations derived from skin dermis, BF is for fibroblast populations derived from bladder lamina propria, and VF is for fibroblast populations derived from vaginal stromae. HI- is for heat-inactivated, U- is for untreated, FBS is for fetal bovine serum, and NBCS is for newborn calf serum. N = 3 independent fibroblast populations for each organ, *n* = 4 tissues (three pictures have been taken of tissue). Scale bars are 100 µm.

**Figure 5 bioengineering-08-00184-f005:**
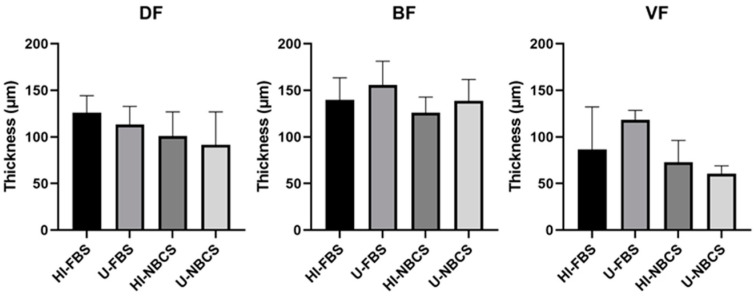
Stroma thickness of reconstructed stromae by fibroblasts primary populations depending on serum type and heat-inactivation status. DF is for fibroblast populations derived from skin dermis, BF is for fibroblast populations derived from bladder lamina propria, and VF is for fibroblast populations derived from vaginal stromae. HI- is for heat-inactivated, U- is for untreated, FBS is for fetal bovine serum, and NBCS is for newborn calf serum. The bars represent the mean ± standard deviation of measurements on a photograph of the tissue slices stained with Masson’s Trichrome protocol. N = 3 independent fibroblast populations for each organ, *n* = 3 tissues (three pictures have been taken of tissue).

**Figure 6 bioengineering-08-00184-f006:**
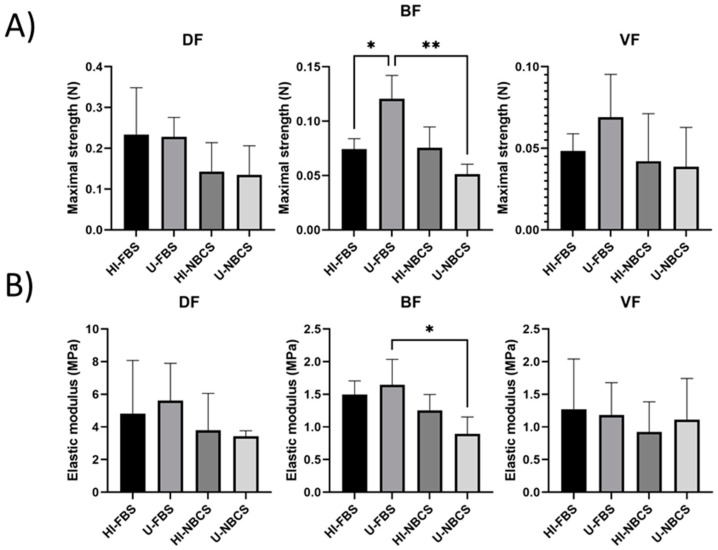
Maximal strength and elastic modulus of reconstructed stromae by fibroblasts primary populations depending on serum type and heat-inactivation status. DF is for fibroblast populations derived from skin dermis, BF is for fibroblast populations derived from bladder lamina propria, and VF is for fibroblast populations derived from vaginal stromae. HI- is for heat-inactivated, U- is for untreated, FBS is for fetal bovine serum, and NBCS is for newborn calf serum. Asterisks (*) illustrate significant differences. One asterisk is for *p*-value between 0.05 and 0.01; two asterisks is for *p*-value between 0.01 and 0.005. (**A**) Maximal strength of reconstructed stromae. The bars represent the mean ± standard deviation of the maximal strength in Newton (N) of the tissues using a uniaxial tensile test. N = 3 independent fibroblast populations for each organ, *n* = 4 tissues. (**B**) Elastic modulus of reconstructed stromae. The bars represent the mean ± standard deviation of the elastic modulus in MegaPascal (MPa) of the tissues using a uniaxial tensile test. The elastic modulus is in inverse relation to elasticity. N = 3 independent fibroblast populations for each organ, *n* = 4 tissues.

**Figure 7 bioengineering-08-00184-f007:**
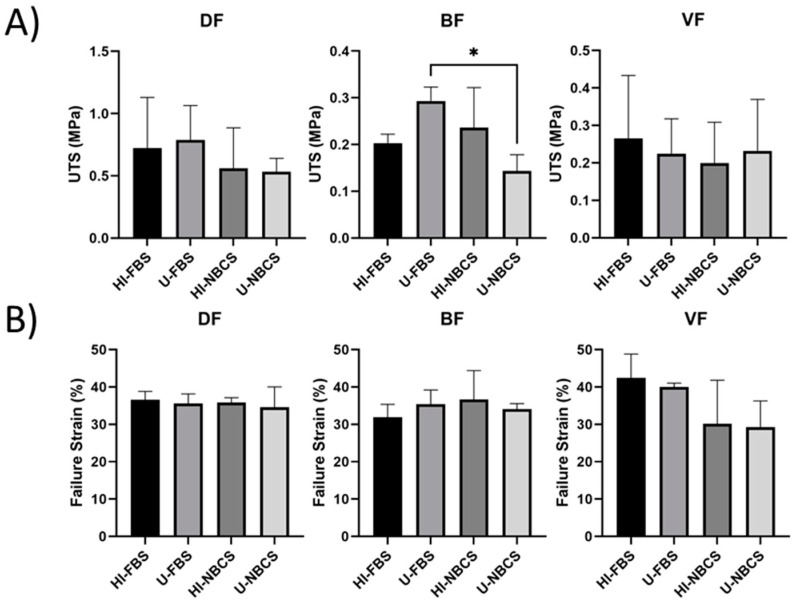
Ultimate tensile strength and failure strain of reconstructed stromae by fibroblasts primary populations depend on serum type and heat-inactivation status. DF is for fibroblast populations derived from skin dermis, BF is for fibroblast populations derived from bladder lamina propria, and VF is for fibroblast populations derived from vaginal stromae. HI- is for heat-inactivated, U- is for untreated, FBS is for fetal bovine serum, and NBCS is for newborn calf serum. Asterisks (*) illustrate significant differences. One asterisk is for a *p*-value between 0.05 and 0.01. (**A**) Ultimate tensile strength of reconstructed stromae. The bars represent the mean ± standard deviation of measurements of the UTS in MegaPascal (MPa) using a uniaxial tensile test. N = 3 independent fibroblast populations for each organ, *n* = 4 tissues. (**B**) Failure strain of reconstructed stromae. The bars represent the mean ± standard deviation of measurements of the failure strain in the percentage of the tissue length (%) using a uniaxial tensile test. N = 3 independent fibroblast populations for each organ, *n* = 4 tissues.

## Data Availability

All data are fully available from the corresponding author on reasonable demand.

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
