# Peer review of "Heat-Inactivation of Fetal and Newborn Sera Did Not Impair the Expansion and Scaffold Engineering Potentials of Fibroblasts"

_bioengineering, 2021, doi:10.3390/bioengineering8110184_

Round 1

Reviewer 1 Report

the authors in this manuscript evaluated  FBS & NBCS in their untreated and heat-inactivated forms for the cell culture of cells from the dermis, bladder and vaginal stroma in both 2D and 3D tissues. This manuscript is of great value to the scientific community, especially for the tissue engineering community, who uses large amount of sera, the question remains 'Is heat-inactivation of sera that is routinely performed, necessary??' 

However, the manuscript can be enhanced by addressing the following points: 

  1. the title should be changed as it does not reflect the work done in 2D and 3D.
  2. In the Keywords: add 2D & 3D cultures
  3. the authors should explain in the discussion why they used cells from three types of tissues (dermis, bladder & vagina))
  4. there are many missing references throughout the study. Add citations to the equations used in lines 118 & 120. 
  5. 5. there is  a lot of  English editing to be done. I will not state all, only a few like line 106 thawed; line 165 (products injected were indicated as ......; line 170; corresponds; line 392 393 why in bold ! 
  6. for the number of cells, it is more suitable to use units cells/cm2 throughout 
  7. figure 6: add scale bars 
  8. Add limitations of this study !! 
  9.  

Reviewer 2 Report

Pellerin et al. provided an interesting study comparing the impact of heat-inactivation of two different sera on fibroblasts from different organs. The results are crucial for all researchers performing cell culture.

The manuscript is generally well written but some points need to be improved, as detailed below:

General comments:

- While in the abstract the conclusion is clear "Overall, serum inactivation does not appear to provide a significant benefit.", in the description of the results in the Results part it is difficult to follow what the authors want to emphasize. The choice of presenting separately the 3 donors results for each DF, BF and VF is surprising and makes the results not enough summarized. The reproducibility of the results in a study is important and the 3 independent experiments need to be merged in the same graph (first make the average of the results in each donor and then show the error bars of the average of the three donors results on the graphs, finally make the statistical analysis on these 3 donors data only). Some assay will lead to no difference at all between the different conditions but it will be more accurate, really representing the actual results obtained and will allow to definitely conclude on the results which is not the case here with many "except for..." or "has variable effects" that are not suitable. The separated data can be put in the supplementary data to highlight a specific outsider donor when necessary.

- Some sentences are unclear and should be modified, examples are given in the detailed comments.

Detailed comments:

-Page 2 line 61 "This concept is interesting because the production is fast and relatively inexpensive.": to temper since it really depends on the biomaterial used.
Next sentence "However, the pre-existing structure inevitably induces bias such as cell misdifferentiation, lack of native scaffold composition and organization, etc.": this sentence is not true and should be removed or improved.

- Materials and Methods part: please add the country and city for all the companies cited.

- Page 3 line 106 "serum bottles were thaw overnight in water at 4°C.": Was it really in water? not directly in the fridge overnight?

- Page 3 line 113 "fibroblasts were seeded at 10% confluence in 12-well plates (2 × 10^4 Fibroblasts).": 2 × 10^4 is the number of fibroblasts per well? in which volume per well?

- Page 3 line 122 "For the adhesion measurement, the cell count of the indicated condition at day one was compared to the cell count at day one for the reference (i.e. heat-inactivated FBS) except for the DF1 population where untreated FBS served as reference.": This sentence is unclear and difficult to understand. And why DF1 has different reference?

- Page 4 line 154 "These measure has been chosen because at this
time point, the OCR or ECAR measurements were stabilized after the indicated product injection.": How to estimate that "measurements were stabilized"? They do not look stabilized on the graphs.

- Page 6 line 219 "This surface is often the plastic of a Petri dish.": is it the case in this study? Which plastic type?

- Figure 2 DF1: Why HI-FBS is at 0%?

- In every graphs captions: "asterisks" should be used instead of "stars"

- Figure 4 and 5: Why the order of the results presented are different from the other figures? with HI-NBCS and U-NBCS being first.
Also on these figures the font size used for the axis is too small.

- Figure 6: The histology images are of too low magnification and does not allow to compare the different conditions. Higher magnification is needed. Also, it would be of interest to compare the cell number/tissue area between the different conditions.
For these results and the next figure, even if several pictures were taken for each sample, only one sample was made from each donor? Reproducibility is poor for these data. 3 replicates for each donor is usually a minimum requirement.

- Figure 8 "Each dot represents a measurement of the maximal strength in Newton (N)": Does it mean that the data here came from repeated measures of the same sample? This is not really reliable and again, 3 tissue replicates for each donor is usually a minimum requirement for accurate analysis.

- Page 17 line 475-480: What are the importance of describing this here? Do you want to explain the various results obtained depending of the donor in the results?

- Examples of unclear sentences:

Page 1 line 31 "Fetal bovine serum (FBS) is currently considered as a reference. However, almost everyone agrees on the need to replace it with less expensive and more ethical sources that would allow more reproducible results [1,2].": the "however" sounds inappropriate, maybe "even if..." would be better

Page 1 line 38 "From the "dark ages" of cell culture,": What the "dark ages" refer to here?

On the first 2 pages, there are too many "nevertheless" used.

Page 2 line 51 "Another application, outside cell culture,": "outside" is not suitable in this sentence

Page 2 line 56 "This set of technologies is similarly based on the use of sera of various origins." : this sentence meaning is confusing

Page 2 line 64-68 : too long sentence

Page 2 line 70-73 : also long sentence and not clear, for instance "it can be expected these reconstructed tissues recreate" would be better using "it can be expected that these...." and what it the meaning of "and those between cells than other tissue engineering models."?

Round 2

Reviewer 2 Report

The modifications performed by the authors really improved the manuscript, especially the figures. The overall message is now easy to understand.

Before publication, to further improve the current version, it would be greatly appreciated that the authors add one or two sentences to discuss the higher maximal strength, elastic modulus and UTS found for the BF U-FBS (Figures 6-7), providing some possible explanations.

Author Response

The modifications performed by the authors really improved the manuscript, especially the figures. The overall message is now easy to understand.
Before publication, to further improve the current version, it would be greatly appreciated that the authors add one or two sentences to discuss the higher maximal strength, elastic modulus and UTS found for the BF U-FBS (Figures 6-7), providing some possible explanations.

We add the following paragraph and references:

Such a result could be, in part, explained by the nature of BFs. In a previous study, the scaffold formed by the bladder fibroblasts showed spots of degraded ECM with increasing MMP activity resulting in a weakened tissue, which is not observed using dermal fibro-blasts [29]. One explanation for such a discrepancy in tissue properties between substitutes derived from bladder and dermal fibroblasts may be the nature of bladder fibroblasts. Several studies illustrate the plasticity of bladder fibroblasts which have a phenotype varying from contractile muscle cells and secretory fibroblasts. High concentrations of Sonic Hedgehog (Shh) induce a fibroblast-like phenotype for bladder fibroblasts that secrete ECM to form the lamina propria and, in their turn, secrete the Bone Morphogenetic Protein 4 (BMP4). This factor, combined with the reduced concentration of Shh along the gradient, induces the smooth-muscle phenotype for the mesenchymal bladder cells in the area of what should become the detrusor [30,31].
29. Bouhout S, Chabaud S, Bolduc S. Organ-specific matrix self-assembled by mesenchymal cells improves the normal urothelial differentiation in vitro. World J Urol. 2016 Jan;34(1):121-30. doi: 10.1007/s00345-015-1596-2. Epub 2015 May 26. PMID: 26008115.
30. Cao M, Tasian G, Wang MH, Liu B, Cunha G, Baskin L. Urothelium-derived Sonic hedgehog promotes mesenchymal proliferation and induces bladder smooth muscle differentiation. Differentiation. 2010 Apr-Jun;79(4-5):244-50. doi: 10.1016/j.diff.2010.02.002. Epub 2010 Mar 15. PMID: 20227816; PMCID: PMC3712847.